# *Trans*- and *Cis*-Phosphorylated Tau Protein: New Pieces of the Puzzle in the Development of Neurofibrillary Tangles in Post-Ischemic Brain Neurodegeneration of the Alzheimer’s Disease-like Type

**DOI:** 10.3390/ijms25063091

**Published:** 2024-03-07

**Authors:** Ryszard Pluta, Stanisław J. Czuczwar

**Affiliations:** Department of Pathophysiology, Medical University of Lublin, 20-090 Lublin, Poland; stanislaw.czuczwar@umlub.pl

**Keywords:** brain ischemia, Alzheimer’s disease, cardiac arrest, tau protein, hyperphosphorylation, *cis*-phosphorylated tau protein, *trans*-phosphorylated tau protein, paired helical filaments, neurofibrillary tangles, genes

## Abstract

Recent evidence indicates that experimental brain ischemia leads to dementia with an Alzheimer’s disease-like type phenotype and genotype. Based on the above evidence, it was hypothesized that brain ischemia may contribute to the development of Alzheimer’s disease. Brain ischemia and Alzheimer’s disease are two diseases characterized by similar changes in the hippocampus that are closely related to memory impairment. Following brain ischemia in animals and humans, the presence of amyloid plaques in the extracellular space and intracellular neurofibrillary tangles was revealed. The phenomenon of tau protein hyperphosphorylation is a similar pathological feature of both post-ischemic brain injury and Alzheimer’s disease. In Alzheimer’s disease, the phosphorylated Thr231 motif in tau protein has two distinct *trans* and *cis* conformations and is the primary site of tau protein phosphorylation in the pre-entanglement cascade and acts as an early precursor of tau protein neuropathology in the form of neurofibrillary tangles. Based on the latest publication, we present a similar mechanism of the formation of neurofibrillary tangles after brain ischemia as in Alzheimer’s disease, established on *trans*- and *cis*-phosphorylation of tau protein, which ultimately influences the development of tauopathy.

## 1. Brain Ischemia

Transient focal brain ischemia in humans is called ischemic stroke and accounts for approximately 70–85% of all cases [1,2]. Stroke is a huge and growing health problem in the modern world, affecting over 40 million cases per year [3]. Currently, it is estimated that the number of post-stroke patients worldwide is approximately 33 million, of which 6 million die, and 5 million remain permanently disabled [1,3,4,5]. It should be noted that the annual cost of treatment and care of stroke patients in Europe is approximately EUR 60 billion [6] and in the USA approximately USD 100 billion [7]. In humans, stroke accelerates the development of dementia by 10 years, and dementia is evident after the first ischemic episode in approximately 10% of patients, and after a recurrent ischemic episode—in as many as over 41% of cases [5,6]. However, after 25 years post-ischemia, the incidence of dementia was estimated at approximately 48% of patients [4,6]. The consequence of this is cognitive impairment in approximately 70% of people who survive stroke [5]. This leads to irreversible changes in the brain and becomes the second cause of dementia, the third cause of disability, and, according to new estimates, it may soon become the main cause of dementia development in the world [5]. Over the last two decades, the average lifetime risk of brain ischemia has increased by 9% [8]. The latest predictions are disturbing, indicating that the number of patients with stroke in the world will increase to 77 million in 2030 [4,5]. If the trend in ischemic brain injury continues, by 2030 approximately 12 million cases will die, 70 million will survive ischemia, and each year there will be disability for over 200 million years [4,6]. Based on the latest clinical and experimental evidence, it was hypothesized that an episode of brain ischemia may contribute to the development of Alzheimer’s disease [9,10,11,12,13,14,15,16,17,18,19,20]. Growing evidence shows that brain ischemia causes neurodegeneration of the Alzheimer’s disease-like phenotype and genotype, and provides new insight into the similar mechanisms of changes that may be involved in the development of both diseases, but the ultimate answer underlying their co-development remains unknown [17,18,21].

## 2. Alzheimer’s Disease-like Phenotype Post-Ischemia

It has been documented that both cerebral ischemia and Alzheimer’s disease share some risk factors [22]. Post-ischemic neuronal death has been shown to occur as a result of excitotoxicity and the influence of folding proteins such as amyloid and modified tau protein [13,23,24,25]. The examination of brain post-ischemia revealed similar changes in the blood-brain barrier [26], as in the brains of patients with Alzheimer’s disease. It has been shown that vasoconstriction during recirculation [27] is accompanied by the development of cerebral amyloid angiopathy [2], also observed in Alzheimer’s disease. During recirculation, oxidative stress affects the genome, causing DNA damage, neuronal, glial and endothelial cell death, and neurological outcomes [17,18,25,28]. In post-ischemic neurodegeneration, acetylcholine loss was mainly found in the hippocampus [29], similar to the brain in Alzheimer’s disease. After ischemic brain injury, the death of pyramidal neurons in the hippocampus predominates, with the development of atrophy of the hippocampus and other brain structures [9], similar to Alzheimer’s disease. Neuroinflammation also plays a major role in the progression of post-ischemic neurodegeneration [30,31,32], as in Alzheimer’s disease.

## 3. Alzheimer’s Disease-like Genotype Post-Ischemia

Studies after experimental brain ischemia have revealed the dysregulation of genes related to the processing of amyloid precursor protein in the CA1 and CA3 subfields of the hippocampus and in the cortex of the temporal lobe (Table 1). Changes in the expression of genes such as *β-secretase* (encoding β-secretase), *presenilin 1* (encoding presenilin 1) and *2* (encoding presenilin 2), and *amyloid precursor protein* (encoding amyloid precursor protein) in CA1 include all genes examined 2, 7, and 30 days post-ischemia (Table 1) [33]. However, changes in the expression of the above genes in CA3 were less severe and did not occur throughout 2–30 days after ischemia and did not involve all genes (Table 1) [34]. In this structure, the above genes were assessed in animals that lived 1, 1.5, and 2 years after ischemia. The changes virtually occurred at all times and were more intense (Table 1) [35]. While changes in gene expression in the temporal cortex immediately occurred post ischemia and did not occur throughout survival and did not involve all genes (Table 1) [36,37]. The data revealed that post-ischemic brain injury in animals and humans induced amyloidogenic processes in the examined brain structures [9,18,25,38,39].

The dysregulation of *BECN1*, *BNIP3*, and *CASP3* genes, associated with neuronal death, has been observed in the post-ischemic neurodegeneration in animals [3]. *BECN1* gene expression in CA1 after brain ischemia with survival at 2, 7, and 30 days was within the control limits (Table 1) [40]. *BNIP3* gene expression 2 days post-ischemia in the CA1 was above the control value, and on days 7–30 following ischemic injury, it was within the control range (Table 1) [40]. *CASP3* gene overexpression in CA1 occurred 2 and 7 days after ischemia, while on day 30 it was below the control value (Table 1) [40].

*BECN1* gene expression in the CA3 area oscillated around the control values after 2 days, was decreased on day 7, and significantly increased on day 30 following ischemia (Table 1) [41]. Post-ischemic *BNIP3* gene expression was below control values for 2–30 days of post-ischemic follow-up (Table 1) [41]. *CASP3* gene tests showed at 2-day decline below control values and increased activity on days 7 and 30 after ischemia (Table 1) [41].

*BECN1* gene expression in the temporal cortex 2 days after ischemia was above the control value, but after 7–30 days it was around control values (Table 1) [42]. Two days after ischemia, *BNIP3* gene expression was below the control value, and after 7–30 days its expression significantly increased (Table 1) [42]. *CASP3* gene expression in the temporal cortex 2 days after ischemia was below the control value, but on days 7–30 its expression was significant (Table 1) [42].

Studies have shown opposite changes in *MAPT* gene expression in CA1 and CA3 areas 2, 7, and 30 days after ischemia (Table 1). In CA1, *MAPT* gene expression significantly increased 2 days after ischemia (Table 1) [43]. But the expression of this gene 7–30 days after the ischemic injury oscillated around the control value (Table 1) [43]. In CA3, fluctuations in *MAPT* gene expression were observed within control limits 2 days post-ischemia (Table 1) [34]. However, a significant increase in *MAPT* gene expression was observed 7–30 days after ischemia (Table 1) [34]. In the CA3 area after ischemia with survival of 1, 1.5, and 2 years, overexpression of the *MAPT* gene was significant (Table 1) [35].

After transient brain ischemia in animals and humans with recirculation evidence shows that Alzheimer’s disease-like folding proteins, amyloid, and tau protein are induced leading to the development of amyloid plaques and neurofibrillary tangles [9,38,39,44,45,46,47].

## 4. Tau Protein and Brain Ischemia

Post-ischemic brain injury in animals and humans has many neuropathological similarities to Alzheimer’s disease, among which the hyperphosphorylation of tau protein is of key importance [25,46,48,49,50,51]. Tau protein is a phosphoprotein that stabilizes the structure of microtubules. The hyperphosphorylation of tau protein causes tau protein and microtubules to lose normal functions [52,53,54]. Hyperphosphorylation post-ischemia causes the mislocalization of tau protein, which contributes to the formation of neurofibrillary tangles in humans and animals [44,46,47,48,49,50,51]. Neurofibrillary tangles are associated with neuronal death and cognitive deficits in Alzheimer’s disease [55]. Hyperphosphorylation also contributes to neuronal apoptosis after brain ischemia [46,56,57,58] whereas its deletion reduces the infarct volume [59]. Based on the above, it appears that tau protein hyperphosphorylation after cerebral ischemia follows a similar pattern of tau protein pathology as in Alzheimer’s disease.

Misfolded and aggregated forms of tau protein produce pathological structures in a number of neurodegenerative diseases, including Alzheimer’s disease. A wealth of evidence has emerged within the last decade to suggest that the misfolded tau protein in tauopathies possesses prion-like features [60,61]. The prion-like concept for tauopathies initially arose from the observation that the progressive accumulation of tau protein pathology as the symptoms of Alzheimer’s disease progress seemed to follow anatomically linked pathways [60,61,62]. Subsequent studies in cell and animal models revealed that misfolded tau protein can propagate from cell to cell, and region to region, in the brain through direct neuroanatomical connections. Studies in cells and mouse models have demonstrated that experimentally propagated forms of misfolded tau protein can exist as conformationally distinct “strains” with unique biochemical, morphological, and neuropathological characteristics [60].

Therefore, one can imagine a kind of “species barrier” in which seeding only occurs when there is a structural similarity between the seed and the template. To support this point, one study has demonstrated seed and template effects on the seeding rate and extent [60]. This specificity of isoform aggregation in an environment in which all isoforms are present further supports the idea that the molecular specificity of the seed and template may be critical determinants in the acceptability of polymer elongation leading to the formation of tau protein inclusions [60].

Alzheimer’s disease is associated with the aggregation of the pathogenic tau protein and prion-like seeding fibril growth in brain cells. The tau protein pathology appears in Alzheimer’s disease when the tau protein transitions into amyloid fibrils, which can spread from cell to cell in a prion-like manner [60,61,62]. Many amyloid proteins, including tau protein and amyloid, aggregate into “cross-β’’ filaments featuring β-sheets that run the length of the fibrils, stabilized by steric zippers. Steric zippers are paired β-sheets mated by tightly interdigitated side chains [63]. Most zipper interfaces exclude water molecules, contributing to filament stability [64]. The development and spread of amyloid fibrils within the brain correlates with disease onset and progression.

## 5. Development of Post-Ischemic Tauopathy

For decades, the importance of tau protein phosphorylation and its role in the pathology of Alzheimer’s disease was unclear. In Alzheimer’s disease, the hyperphosphorylation of tau protein occurs at various residues [53]. The Thr231 residue is the main phosphorylation site of tau protein, appearing in the process of its phosphorylation before the development of neurofibrillary tangles, and its level and severity over time are related to the progression of Alzheimer’s disease [65,66]. Phosphorylation of the Thr231 residue in tau protein occurs in two different conformations, *trans* and *cis* [66]. Thus, using antibodies capable of recognizing two conformations, it was shown that *trans*-phosphorylated Thr231-tau protein has physiological properties, while *cis*-phosphorylated Thr231-tau protein exhibits pathological properties [67]. Peptidylprolyl *cis*/*trans* isomerase has been shown to convert the *cis* to *trans* conformation of phosphorylated-tau protein, thereby preventing tau protein pathology in Alzheimer’s disease [66]. *Trans-*phosphorylated tau protein is associated with microtubules, while *cis-*phosphorylated tau protein, which is resistant to dephosphorylation, acts as an early precursor of tau protein pathological processes, for example, in post-traumatic encephalopathy and Alzheimer’s disease [66,67]. According to research in clinical and preclinical traumatic brain injury, neuronal cells produce *cis*-phosphorylated tau protein a long time before tau protein aggregates, which spreads to neighboring neurons and ultimately leads to their apoptosis [67,68]. Immunotherapy against the *cis*-phosphorylated tau protein has been shown to effectively work in preclinical models of brain injury by preventing neuronal cell death [67,69].

Increased tau protein immunoreactivity in neuronal and neuroglial cells, mainly in the hippocampus, brain cortex, and thalamus, has been demonstrated in animals and also in humans post-ischemia [48,49,70,71,72,73,74,75,76,77,78]. Data indicate that neuronal and neuroglial cells store tau protein during and after experimental ischemia [73], which indicates the progression of pathological phenomena in the ischemic brain related to tau protein [76]. It was found that the altered tau protein hinders the movement of organelles, neurofilaments, and amyloid precursor protein, and increases oxidative stress in neurons, causing the accumulation of amyloid precursor protein in their cytoplasm [79]. Moreover, brain microdialysis after experimental ischemia showed an increase in the level of total tau protein in the brain [80]. Specific kinases that are involved in the hyperphosphorylation of tau protein after ischemia, such as CDK5 (cyclin-dependent kinase 5), GSK3β (glycogen synthase kinase-3 beta), ERK (extracellular signal-regulated kinase), protein phosphatase 2, and asparagine endopeptidase, are activated [46,48,49,58,81,82]. Subsequently, data from the study by Mankhong et al. [83] revealed increased acetylation and nitrosylation of tau protein after experimental focal brain ischemia with a simultaneous impact on the intensification of neurodegenerative processes post-ischemia. The importance of acetylation of tau protein was confirmed in a study showing that reducing acetylated tau protein has a neuroprotective effect in traumatic brain injury in animals and humans [84]. This indicates that tau protein is an important element in the development of neurofibrillary tangles after ischemic brain injury in humans [44,47,84]. Tauopathy, which, as we can see, also refers to post-ischemic brain damage, is associated with post-translational changes in the tau protein through acetylation, ubiquitination, and phosphorylation, which disrupts the structure of microtubules and reduces the solubility of the tau protein and promotes its aggregation [85]. It is important to note that tau protein acetylation has been identified as one of the important factors in brain neurodegeneration [86,87]. Tau protein acetylation inhibits its degradation and contributes to the development of tauopathies and is a critical determinant of the modulation of tau protein aggregation and clearance [86,87]. Studies conducted by Mankhong et al. [83] showed a long-term increase in tau protein acetylation in the cerebral cortex after experimental ischemia. These studies also showed that hyperacetylation of tau protein caused a reduction in sirtuin 1 activity [83]. These observations are consistent with clinical studies that propose tau protein acetylation as a new diagnostic biomarker in patients with Alzheimer’s disease [87].

The latest work reveals the induction of *cis*-phosphorylated tau protein after experimental cerebral ischemia [88]. The article probably presents the most important elements in the development of tau protein pathology after brain ischemia, namely the presence of *trans*- and *cis*-phosphorylated tau protein [88]. Cerebral ischemia has been shown to induce the development of *cis*-phosphorylated-tau protein and its gradual accumulation over time, which contributes to the development of cistauosis and progressive post-ischemic neurodegeneration [88]. *Cis-*phosphorylated-tau protein is involved in neuronal cell death under these conditions, which has been documented by the positive effect of immunotherapy directed against *cis*-phosphorylated-tau protein [67].

The modification of tau protein by hyperphosphorylation is one of the hallmarks of Alzheimer’s disease, which is present in post-ischemic brain damage [13,25,51,56,57,58,89]. Studies have indicated that hyperphosphorylation of tau protein has close pathological links with excitotoxicity, mitochondrial dysfunction, oxidative stress, autophagy, apoptosis, neuroinflammation, and changes in the blood-brain barrier in the development of post-ischemic neurodegeneration [13,23,25,26,30,31,32,90,91,92]. Namely, excitotoxicity, oxidative stress, and blood-brain barrier failure, which are enhanced by tau protein hyperphosphorylation, contribute to tau protein phosphorylation in a vicious circle pattern after ischemia [23,26,91,93,94]. Studies have revealed that the hyperphosphorylation of tau protein induced by brain ischemia leads to the development of paired helical filaments [51], neurofibrillary tangles-like [46] and neurofibrillary tangles (Figure 1) [44,47]. Additionally, studies have revealed that tau protein deficiency has a neuroprotective effect in models of brain ischemia [91,95]. Thus, these studies demonstrate the importance of tau protein in the neuropathology of brain ischemia in animals and humans, but also point to similar abnormalities of tau protein in the brain after ischemia and Alzheimer’s disease. In this way, post-ischemic neurodegenerative processes with the Alzheimer’s disease-like phenotype can be at least partially explained. In brain ischemia, tau protein is phosphorylated at many sites [53,56,57], as in Alzheimer’s disease. It is also an early indicator of developing Alzheimer’s disease in the cerebrospinal fluid [96]. An increased level of Thr231-phosphorylated tau protein in patient cerebrospinal fluid is associated with cognitive impairment, the accumulation of neurofibrillary tangles in the brain, and hippocampal atrophy, indicating progression from mild cognitive impairment to Alzheimer’s disease [67,69,96].

According to research in Alzheimer’s disease, the phosphorylated Thr231 motif of tau protein tends to undergo *trans* to *cis* isomerization, leading to increased levels of toxic *cis-*phosphorylated tau protein [66,67]. It has also been shown that the ischemia-induced phosphorylated Thr231-tau protein also has a *cis* conformation [88]. This indicates that *cis*-phosphorylated-tau protein plays a key role in tau protein neurotoxicity following brain ischemia. Both Alzheimer’s disease and cerebral ischemia induce the activation of death-associated protein kinase 1. In this situation, activated death-associated protein kinase 1 phosphorylates tau protein and triggers pathological changes in neuronal cells [97,98,99,100]. The death-associated protein kinase 1 regulation of tau protein properties is mediated by its phosphorylation of peptidyl-prolyl *cis/trans* isomerase, which inactivates peptidyl-prolyl *cis/trans* isomerase, leading to increased levels of toxic *cis*-phosphorylated-tau protein [97,100]. Thus, research has indicated that cerebral ischemia triggers the activation of death-associated protein kinase 1 [99] and suggests that this contributes to the increase in *cis*-phosphorylated tau protein levels after cerebral ischemia. These observations support the concept that *cis*-phosphorylated-tau protein influences post-ischemic neuropathogenesis and is likely one of the earliest steps involved in ischemic neuronal cell changes and death.

A mouse model of neurodegeneration has shown that *cis*-phosphorylated-tau protein is not only limited to the cerebral cortex but also spreads to other structures such as the hippocampus within six months after injury [67], which has been confirmed in patients with traumatic encephalopathy [68]. Therefore, it is proposed that *cis*-phosphorylated-tau protein may contribute to the progression of secondary neurodegeneration in various brain regions of the primary infarct within days to months of its onset and may be a target for the treatment of transient cerebral ischemia [88,101]. Importantly, studies in mice have revealed that the use of monoclonal antibodies against *cis*-phosphorylated tau protein effectively prevented cistauosis by reducing intra- and extracellular levels of *cis*-phosphorylated tau protein, inhibiting its spread to neighboring neurons and stopping neuronal cell death [67,68]. Thus, these data not only represent a new molecular process during post-ischemic brain neurodegeneration, but also, considering previous studies on immunotherapy against *cis*-phosphorylated-tau protein in experimental brain injury [67,68], propose an effective therapeutic agent to inhibit cistauosis and neurodegeneration in the post-ischemic brain.

## 6. Conclusions

Brain ischemia contributes to the development of brain neurodegeneration as a result of various processes similar to those that occur in Alzheimer’s disease. Cerebral ischemia-reperfusion causes a genotype and phenotype similar to Alzheimer’s disease. Post-ischemic injury affects, among others, *MAPT* gene expression, tau protein acetylation, the activation of many kinases closely related to tau protein hyperphosphorylation, such as GSK3β, CDK5, and ERK, the development of *cis*-phosphorylated-tau protein, paired helical filaments, and, finally, neurofibrillary tangles. These changes lead to the development of post-ischemic tauopaty. The demonstration of the *cis*-phosphorylation of tau protein after ischemia shows, for the first time, how the pathological form of tau protein can spread to neighboring ischemic neurons. This is probably one of the most important recent observations, which shows a possible mechanism for the spread of tau protein pathology from neuron to neuron after cerebral ischemia. Research on the presented *cis*-phosphorylated tau protein is an important reference point for future studies on post-ischemic tauopathy, which may contribute to the early detection and effective treatment of this lesion.

## Figures and Tables

**Figure 1 ijms-25-03091-f001:**
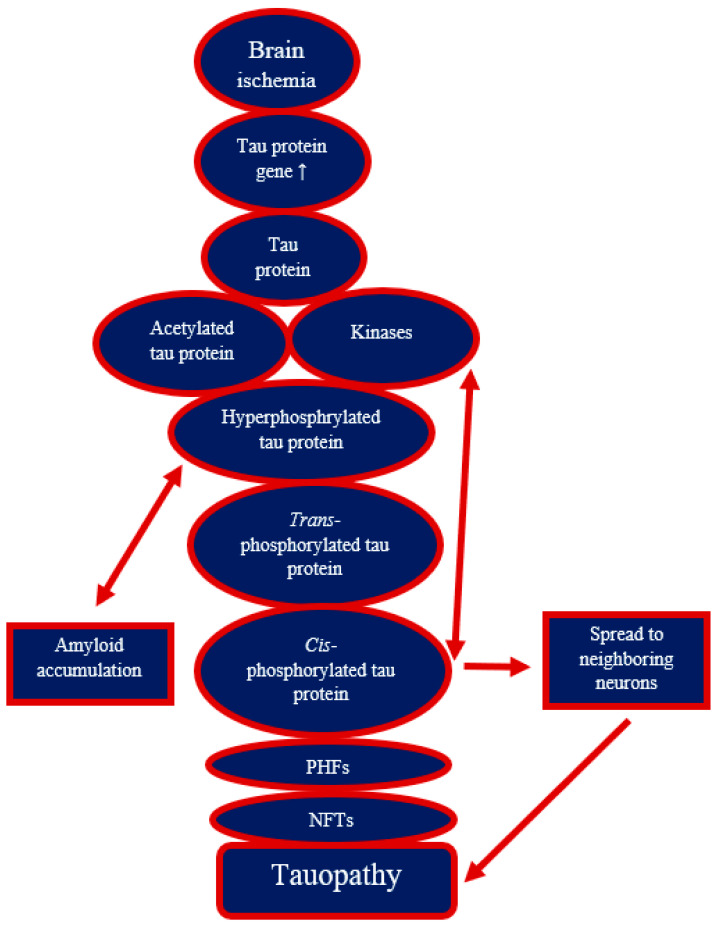
Development of neurofibrillary tangles and tauopathy after brain ischemia. ↑—increase; PHFs—paired helical filaments; NFTs—neurofibrillary tangles.

**Table 1 ijms-25-03091-t001:** Changes in the expression of Alzheimer’s disease-related genes after experimental brain ischemia in the hippocampus and temporal lobe, which are particularly vulnerable to ischemia and among the first to show abnormalities in the early stages of Alzheimer’s disease and play a key role in the control of learning and cognitive functions.

Genes Survival	*APP*	*ADAM10*	*BACE1*	*PSEN1*	*PSEN2*	*BCEN1*	*BNIP3*	*CASP3*	*MAPT*
Selectively vulnerable hippocampal CA1 subfield to ischemia									
2 days	↑	N.A.	↑	↑	↑	↔	↑	↑	↑
7 days	↑	N.A.	↑	↑	↑	↔	↔	↑	↔
30 days	↑	N.A.	↓	↓	↓	↔	↔	↓	↔
Less sensitive hippocampal CA3 subfield to ischemia									
2 days	↔	↓	↓	↑	↔	↔	↓	↓	↔
7 days	↑	↓	↓	↑	↓	↓	↓	↑	↑
30 days	↔	↓	↑	↔	↑	↑	↓	↑	↑
1 year	↑	↑	↑	↑	↑	N.A.	N.A.	N.A.	↑
1.5 year	↑	↑	↓	↔	↔	N.A.	N.A.	N.A.	↑
2 years	↑	↑	↑	↑	↑	N.A.	N.A.	N.A.	↑
Medial temporal lobe cortex									
2 days	↓	N.A.	↑	↔	↑	↑	↓	↓	N.A.
7 days	↑	N.A.	↔	↔	↔	↔	↑	↑	N.A.
30 days	↑	N.A.	↔	↔	↔	↔	↑	↑	N.A.

Expression: ↑—increase; ↓—decrease; ↔—oscillation around control values; N.A.—not available. Genes: *APP*, (encoding amyloid precursor protein); *ADAM10*, (encoding disintegrin and metalloproteinase domain-containing protein 10, which has α-secretase activity); *BACE1*, (encoding β-secretase); *PSEN1*, (encoding presenilin 1); *PSEN2*, (encoding presenilin 2); *MAPT*, (encoding tau protein); *BECN1*, (encoding beclin-1 playing a critical role in the regulation on of autophagy); *BNIP3,* (encoding BCL2/adenovirus E1B 19 kDa protein-interacting protein 3, which has a role in mitophagy); *CASP3*, (encoding caspase 3, which has role in apoptosis).

## Data Availability

The data presented in this study are available on request from the corresponding author.

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
