# Peer review of "Trans- and Cis-Phosphorylated Tau Protein: New Pieces of the Puzzle in the Development of Neurofibrillary Tangles in Post-Ischemic Brain Neurodegeneration of the Alzheimer’s Disease-like Type"

_ijms, 2024, doi:10.3390/ijms25063091_

Round 1

Reviewer 1 Report

Comments and Suggestions for Authors

See PDF file.

Author Response

Reviwer 1.

All changes are in red.

Trans- and cis-phosphorylated tau protein: new pieces of the puzzle in the development of neurofibrillary tangles in post-ischemic brain neurodegeneretion of the Alzheimer's disease type

Manuscript ID: ijms-2889046

Major reminders:

The authors previously hypothesized that brain ischemia is associated with increased neurotoxicity of amyloid beta and tau, which may initiate and develop Alzheimer's disease (AD)-type neurodegeneration. According to this hypothesis, brain ischemia and ischemic dysfunction of the blood-brain barrier can be a trigger for tauopathy (Pluta et al. J Alzheimers Dis. 2018;66(2):429-437) as a common pathological feature of post-ischemic neurodegenerative processes (brain injury) with the AD phenotype. In this manuscript, the authors repeat the information from their previous reviews supplemented by the finding that cis-phosphorylated tau may be the main neurotoxic conformer of tau. The goal/contribution of the manuscript can therefore be the specification of the hypothesis that brain ischemia causes tauopathy associated with the neurotoxicity of cis-phosphorylated tau.

We do not agree with the reviewer's statement that this manuscript repeats previous publications. The main goal of this work is to show step by step the formation of neurofibrillary tangles after cerebral ischemia in humans and animals, based on the latest data.

If this is indeed the intention of the authors, then they should revise the manuscript in such a way that they both clearly declare their hypothesis and objectives, and that they more clearly and fully summarize and discuss the results supporting this hypothesis. Finally, it should be emphasized that this is a hypothesis based mainly on observations on model animals, in the case of cis-neurotoxicity mainly on cultured neurons.

We would like to inform You once again that “the main goal of this work is to show step by step the formation of neurofibrillary tangles after cerebral ischemia in humans and animals, based on the latest data ((see, among others, MS title)”. The manuscript does not focus on cis-neurotoxicity. The data we present are from studies of cerebral ischemia in humans and animals, not from in vitro studies.

It should also be stated and discussed that the initiation of neurodegeneration of the Alzheimer's disease-type may not be identical to the initiation of Alzheimer's disease as such, i.e., that brain ischemia-induced initiation and progress of neurodegeneration of the AD-type means that brain ischemia is only a risk factor for AD.

We cannot fully agree with the reviewer's statement, numerous data and a large group of scientists indicate ischemic etiology of Alzheimer's disease (see citations in MS numbers 10, 11, 12, 13, 14, 15, 16, 17, 19, 20, 21 and others in PubMed not quoted). This data cannot be ignored, as over 100 years of research on Alzheimer's disease has not yet clarified its etiology. There is a very high probability that silent asymptomatic cerebral ischemic episodes trigger Alzheimer's disease, as evidenced by, for example, the appearance of clinical symptoms in AD approximately 20 years after its onset. Additionally, cerebral ischemia and Alzheimer's disease are two diseases that primarily damage the hippocampus; there are no other diseases that cause this type of changes.

It could also be debated to what extent cerebral ischemia may be a consequence of already developing AD.

See explanation above. The question then becomes how or what would initially trigger ischemia by AD? Currently, it is known that an amyloid- and tau protein-based etiology has not been proven. It doesn't just happen. With this approach, we are at a dead end. It seems to us that we cannot block another approach to explaining AD in the absence of it, it would be unethical from the point of view of patients. Moreover, the article does not address this issue and focuses on the mechanisms of formation and presence of NFTs after ischemia.

The authors try to connect several AD hypotheses, mainly the tau hypothesis, the amyloid hypothesis and the neurovascular hypothesis.

We do not do this, we point out that ischemia triggers the formation of amyloid and changes in the tau protein, i.e. it initiates changes in these two molecules.

They do so only very generally and incompletely, e.g., they do not mention at all the neurotoxicity of amyloid beta oligomers and tau oligomers and they do not deal much with the mechanisms of spread of tau pathology in the brain.

We note the neurotoxicity of amyloid and tau protein oligomers in the original version of the manuscript (e.g., see Figure 1) and are now expanding to include the pathology of tau protein spread , although this is not the main focus of the manuscript.

The association of AD with cerebral ischemia is documented in more detail only by changes in the expression of genes involved in the pathophysiology of both AD and ischemia and other neurodegenerative diseases, but this is a repetition of data from the authors' earlier publication [18].

The links between ischemia and Alzheimer's disease concern not only gene changes, as the reviewer claims, but also changes in amyloid (see reference 9,18,38,39,45,77,79) and tau protein (see reference 44,46,47,48,49, 50,51,56,5758,70,71,72,73,74,75,76, 77,78,79,80, 83,84,88,9195) after ischemia and is not a complete repetition of work 18. In this context, we replaced work 18 with the latest work on genes transporting amyloid and tau protein after ischemia.

If changes in gene expression are to confirm the association between cerebral ischemia and AD, then the production of apolipoprotein E4 (ApoE4) should also be discussed, because the strongest genetic risk factor for sporadic AD is the presence of APOE ε4 allele. ?????

We are aware of this problem and are currently conducting research on changes in the expression of apolipoproteins E, J and A1 genes after various survival times after ischemia, from 2 days to 2 years.

The connection of the non-specific pathophysiology after cerebral ischemia with the specific pathophysiology of AD should be described in much more detail and more completely. In addition, it should be clearly stated whether the evidence presented relates to measurements in animal models or humans with AD.

We have already mentioned this problem above, we have expanded the descriptions to indicate whether the studies were conducted on animals or after ischemia in humans. The main contribution of the presented manuscript can be considered the reminder of the higher occurrence and neurotoxicity of the cis-conformation of phosphorylated tau compared to the trans-conformation of phosphorylated tau at Thr231 in AD. But this was already done by the authors of original works dealing with this issue (Nakamura et al. Cell. 2012;149(1): 232-44), including therapeutic implications in AD ([53,55], Nakamura et al. Curr Mol Med. 2013;13(7):1098-109).

To support our line of thinking, the manuscript cites the work of Nakamura et al. from 2012, which is probably not prohibited and proves our extensive knowledge on the presented topic.

This review does not introduce any innovative approach or analysis of known data; at least a discussion of what molecular mechanism may link ischemic stroke with dementia of the AD phenotype would be beneficial.

We do not present a "review" type of work, but a "perspective". Completely new information in the work cited as 88 of 2024 is the description of the occurrence of trans- and cis-phosphorylated tau protein, a previously missing link in the formation of NFTs after ischemia. As new data emerge, we present discussions on possible molecular mechanisms that may link ischemic stroke with dementia with the AD phenotype (step-by-step development of NFTs and their role, especially after ischemia).

Minor reminders:

Full text:

  • Use shorter sentences and more frequent paragraphs. Too long sentences are often used, with inappropriate sentence structure and difficult comprehensibility.

Done.

  • Cite more original work rather than reviews.

15 original works added. Among the added original works there are 10 of our works, we are afraid that if the work is accepted for publication, the editors will ask to remove them as self-citations?!

  • Some statements/reasonings are repeated several times.

We removed repetitions.

Abstract: It should already be stated in the abstract that it is a hypothesis and in the introduction this hypothesis should be clearly and concisely formulated.

Done.

Lines 12, 62, 71: Replace “identical” e.g., with “similar”. I think that at the level of cells and tissues, changes and mechanisms in AD and brain ischemia cannot be called "identical".

Done.

Line 39: A more recent figure should be given on the annual cost of treating and caring for patients after brain ischemia.

Done.

Lines 33 to 58: The need for research into neurodegenerative diseases is justified at great length. A few sentences are all it takes.

Done.

Line 68: Correct “share the same risk factors” to “share some risk factors”.

Done.

Line 88: Only write gene names in italics. β-secretase, presenilin 1 and 2, and amyloid protein precursor are not gene names but protein names. Correct “Changes in the expression of genes such as β-secretase, presenilin 1 and 2, and amyloid protein precursor” to “Changes in the expression of genes such as BACE1 (encoding β-secretase), PSEN1 (encoding presenilin 1), PSEN2 (encoding presenilin 2), and APP (encoding amyloid precursor protein)”.

In the original version of the manuscript, the genes were in italic, but after the editors inserted MS in templete, italic disappeared, sorry. Thank you for your important comment about genes. Done.

Lines 98 and 99: Gene names should be in italics. Gene description is incorrect, e.g. instead of “ADAM10, α-secretase” should be “ADAM10, encoding disintegrin and metalloproteinase domain-containing protein 10, which has α-secretase activity”, instead of “BECN1, autophagy” should be “BECN1, encoding beclin-1 playing a critical role in the regulation on of autophagy”, instead of BNIP3, mitophagy” should be “BNIP3, encoding BCL2/adenovirus E1B 19 kDa protein-interacting protein 3, which has a role in mitophagy”.

Thank you for your important comment about genes. Done.

Line 96: In Table 1 it is not stated that this is a measurement in laboratory animals.

Done.

Lines 101 and 102: Keep only the gene names in italics.

Thank you for your important comment about genes. Done.

Lines 84 to 126: This is a repetition of data from publicationon [18]. If Table 1 and its description are only intended to show the regulation of gene expression after ischemia in genes involved in the pathophysiology of AD, then it is sufficient to write it more concisely and clearly.

We have cited original works and added new data.

Lines 134-150: In the chapter "Tau protein and brain ischemia" there is no mention of tau oligomers and bioactive tau seeds in AD.

Done.

Line 183: Abbreviations CDK5, GSK3β, and ERK are unexplained.

Done.

Lines 203 and 208: It is not customary to cite the title of the journal.

Dane.

Lines 215-240: Separate information about phosphorylated tau and cis forms of phosphorylated tau into paragraphs or subsections.

Done.

Chapter 5.: The chapter needs to be revised - it is confusing, with repetitive information and bad wording.

Done.

Figure 1.: The relationship between “Amyloid accumulation” and “Hyperphosphorylated tau protein” is correctly represented by a double-sided arrow; the one-way arrow from “Amyloid accumulation” to “Tauopathy” should be deleted.

Thanks. Done.

Conclusions: Conclusions are written unconvincingly. They should focus on the main goal/hypothesis of the manuscript.

Changed.

Reviewer 2 Report

Comments and Suggestions for Authors

Sorry about being harsh, but as a paper focusing on tau, the subject matter is interesting, but there are many explanations of events that seem to have little relevance. I feel that it lacks cohesiveness, and a more focused approach would be better.

Perhaps it is a problem with the English language, but the authors’ arguments are subjectively drawn out, and there are many areas that should be described more objectively and without misunderstanding. It would be better to add sufficient explanations that readers with no knowledge in this field can understand. Explanations of the figures and diagrams are also difficult to understand, and we recommend reconsideration.

The titles of the chapters are somewhat rambling and we recommend re-examining them. 2.

2. Alzheimer's disease phenotype post-ischemia" would be more appropriate, for example, "Alzheimer's disease-like phenotype post-ischemia".

Isn't there a difference between AD and AD-like?

3.Alzheimer's disease genotype post-ischemia 

This is also a confusing expression: if it is about AD risk genes involved in the disease, this title would be appropriate, but is it not actually a description of gene expression?

It should be "Expression change of proteins associated with AD in post-ischemia" or something similar.

The description in Table 1 lacks specificity and is poorly understood. What does Genetic study refer to? It should be explained in the text alone so that it is clear what kind of model (or patient?) was studied. It is also good to explain one's own thesis, but are there any other relevant papers? would not it be more perspective to present the reader with a summary of past and present information and show the advanced nature of his research? Table 1 shows the changes in each region of some models, but the region names are unevenly arranged, difficult to understand, and should be corrected. The legend should be rewritten in detail. Table 1 contains many simple facts, and it is not clear whether they support the claims of this text, so they should explain what events are associated with their issue. How does Table 1 relate to this story by focusing on the cis form of tau?

As mentioned above, many expressions equate ischemia and AD.

I do not deny that there are many similarities, but there are many expressions that seem to completely equate the two and confuse the reader, so it would be better to revise them accordingly.

I would mention Line101-103 as typical, but the review of 3 you cite focuses on cerebral ischemia, which is not really related to AD, is it? I do not understand why AD was mentioned in this sentence. If they are related to AD, more specific papers should be cited.

I guess that the cis and trans types are not clear to the reader just by reading the text.

I think it would be better to illustrate this point. In addition, the discussion of other post-translational modifications, such as acetylation, starts out of nowhere, but I am not sure how it relates to the discussion of cis-forms; therefore, if it is related, the introduction should be better. Is this a topic necessary?

Figure 1 is unclear. If the authors believe that other post-translational modifications or events are involved in cis-typing, they should clearly state that in the text. Personally, I think it is too suggestive to include things for which no direct link has been shown. The multifactorial writing style seemed distracting. In addition, the font is difficult to read; therefore, it might be better not to black out the text.

Author Response

Review 2.

All changes are in red.

Sorry about being harsh, but as a paper focusing on tau, the subject matter is interesting, but there are many explanations of events that seem to have little relevance. I feel that it lacks cohesiveness, and a more focused approach would be better.

Done.

Perhaps it is a problem with the English language, but the authors’ arguments are subjectively drawn out, and there are many areas that should be described more objectively and without misunderstanding. It would be better to add sufficient explanations that readers with no knowledge in this field can understand. Explanations of the figures and diagrams are also difficult to understand, and we recommend reconsideration.

Done.

The titles of the chapters are somewhat rambling and we recommend re-examining them. Done.

  1. Alzheimer's disease phenotype post-ischemia" would be more appropriate, for example, "Alzheimer's disease-like phenotype post-ischemia".

Changed.

Isn't there a difference between AD and AD-like?

Based on current knowledge on this topic, no for us.

3.Alzheimer's disease genotype post-ischemia 

This is also a confusing expression: if it is about AD risk genes involved in the disease, this title would be appropriate, but is it not actually a description of gene expression?

Changed for Alzheimer's disease-like genotype post-ischemia.

It should be "Expression? change of proteins associated with AD in post-ischemia" or something similar.

Correct, there is no protein expression, there may be protein level, we stick to the above change. 

The description in Table 1 lacks specificity and is poorly understood.

Changed .

What does Genetic study refer to?

For ischemic model! Done.

It should be explained in the text alone so that it is clear what kind of model (or patient?) was studied.

Ischemic model of AD. Done.

It is also good to explain one's own thesis, but are there any other relevant papers? would not it be more perspective to present the reader with a summary of past and present information and show the advanced nature of his research?

No relevant papers on gene changes. There are only our works, we have added 10 of our works on genes after ischemia. The paper also presents changes in amyloid (see reference 9,18,38,39,45,77,79) and tau protein (see reference 44,46,47,48,49, 50,51,56,5758,70,71,72,73,74, 75,76,77,78,79,80, 83,84,88,9195) after ischemia.

Table 1 shows the changes in each region of some models, but the region names are unevenly arranged, difficult to understand, and should be corrected.

It is experimental ischemic model of AD. Descriptions of brain structures have been improved.

The legend should be rewritten in detail.

Done.

Table 1 contains many simple facts, and it is not clear whether they support the claims of this text, so they should explain what events are associated with their issue.

What is associated with the regions is highlighted in the title of the table: learning and cognition.

How does Table 1 relate to this story by focusing on the cis form of tau?

Increased expression of the MAPT gene after ischemia, and this is associated with cis-phosphorylated tau protein and further development of NFTs step by step. 

As mentioned above, many expressions equate ischemia and AD.

Yes.

I do not deny that there are many similarities, but there are many expressions that seem to completely equate the two and confuse the reader, so it would be better to revise them accordingly.

Done.

I would mention Line101-103 as typical, but the review of 3 you cite focuses on cerebral ischemia, which is not really related to AD, is it?

It was changed.

I do not understand why AD was mentioned in this sentence.

AD removed.

If they are related to AD, more specific papers should be cited.

No more specific papers.

I guess that the cis and trans types are not clear to the reader just by reading the text.

Sorry. We have the opposite opinion.

I think it would be better to illustrate this point.

Sorry. We have no idea how to illustrate it in an accessible way so that an ordinary reader can understand it, it is a chemical term?!

In addition, the discussion of other post-translational modifications, such as acetylation, starts out of nowhere, but I am not sure how it relates to the discussion of cis-forms; therefore, if it is related, the introduction should be better. Is this a topic necessary?

This is essential to the overall formation of NFTs as shown step by step in the figure 1 - part of the molecular changes.

Figure 1 is unclear. If the authors believe that other post-translational modifications or events are involved in cis-typing, they should clearly state that in the text.

There are no other factors at this time.

Personally, I think it is too suggestive to include things for which no direct link has been shown.

Sorry. Figure 1 shows the direct connection of many elements in the creation of NFTs.

 The multifactorial writing style seemed distracting.

The factors in MS are essential and show step by step in the development of NFTs post-ischemia.

 In addition, the font is difficult to read; therefore, it might be better not to black out the text. We changed the background to blue.

Round 2

Reviewer 1 Report

Comments and Suggestions for Authors

The work has been greatly improved and is more readable. All my specific minor reminders were accepted, except for the request to mention the neurotoxicity of tau oligomers. Most of my comments marked as major reminders were not accepted by the authors. These comments were aimed at warning that the hypothesis of the ischemic etiology of Alzheimer's disease is still far from being proven, and certainly only applies to a part of Alzheimer's patients. However, one can accept the authors' different approach to how they evaluate the strength of their arguments based on data that support the hypothesis of the ischemic etiology of Alzheimer's disease. The manuscript may be published as is, or with only these very minor modifications:

Lines 74-75: Gene abbreviations are used in Table 1 and its legend, so it would be consistent to use gene abbreviations (BACE1, PSEN1, PSEN2, and APP) also in lines 74 and 75. If you do not want to use gene abbreviations, at least correct "amyloid precursor protein" to "amyloid beta precursor protein".

Chapter 4: Insert mention of neurotoxicity of tau oligomers.

Reviewer 2 Report

Comments and Suggestions for Authors

The problems have been remedied and are now easier to read.

Personally, I would recommend illustrating the difference between cis and trans tau for beginners, as it is difficult to understand from the text, but this is not a serious problem.